# Polyacrylamide-Based Block Copolymer Bearing Pyridine Groups Shows Unexpected Salt-Induced LCST Behavior

**DOI:** 10.3390/molecules28072921

**Published:** 2023-03-24

**Authors:** Yunyun Tu, Dandan Fang, Wanli Zhan, Zengming Wei, Liming Yang, Penghui Shao, Xubiao Luo, Guang Yang

**Affiliations:** 1Key Laboratory of Jiangxi Province for Persistent Pollutants Control and Resources Recycle, Nanchang Hangkong University, Nanchang 330063, China; 2Biomass Molecular Engineering Center, Anhui Agricultural University, Hefei 230036, China

**Keywords:** thermo-responsive, block-copolymer, LCST, self-assembly, nanomaterial

## Abstract

Thermal-responsive block copolymers are a special type of macromolecule that exhibit a wide range of applications in various fields. In this contribution, we report a new type of polyacrylamide-based block copolymer bearing pyridine groups of polyethylene glycol-block-poly(*N*-(2-methylpyridine)-acrylamide; Px) that display distinct salt-induced lower critical solution temperature (LCST) behavior. Unexpectedly, the phase-transition mechanism of the salt-induced LCST behavior of Px block copolymers is different from that of the reported LCST-featured analogues. Moreover, their thermo-responsive behavior can be significantly regulated by several parameters such as salt species and concentration, urea, polymerization degree, polymer concentration and pH values. This unique thermal behavior of pyridine-containing block copolymers provides a new avenue for the fabrication of smart polymer materials with potential applications in biomedicine.

## 1. Introduction

Stimuli-responsive polymers that are able to change their own physical or chemical structures when triggered by external stimuli (temperature, pH, light, chemicals, etc.) have received a remarkable level of attention due to their potential applications in various fields such as drug delivery, biosensors and intelligent device manufacturing [1,2,3,4,5]. Over the past few decades, some stimuli-responsive polymers have been exploited for the construction of smart nanomaterials with different application potentials [6]. Among these, temperature-responsive polymers represent a special class that undergoes a phase transition in solvents at a critical solution temperature (lower critical solution temperature (LCST) or upper critical solution temperature (UCST)) [7]. Generally, LCST-type block copolymers (BCPs) in water are more attractive due to their potential bio-related applications [8,9,10,11,12]. For example, poly(*N*-isopropylacrylamide; PNIPAM)-based BCP is the most popular, with an LCST of around 32 °C—which approaches physiological temperatures [13]. PNIPAM bearing both hydrophilic (amide) and hydrophobic (isopropyl) groups has an LCTS due to the dehydration of its amide group upon heating, leading to a cloudy aqueous solution. Another class of polymers are the poly (oligo(ethylene glycol) (meth)-acrylate))-based BCPs that have an oligo ethylene glycol (OEG) as their temperature-responsive group, which exhibit a tunable LCST with a large range reaching up to 70 °C, depending upon the terminal groups (-Me or -Et) or their repeat units of oligo ethylene glycol [14,15,16].

It has been widely accepted that salts can enable the precipitation of certain proteins from aqueous solutions, which follows a recurring trend known as the Hofmeister series [17,18,19]. There is a general phenomenon that anions can cause this behavior to be more pronounced than cations can. The represented order of the anion species is: CO_3_^2−^ > SO_4_^2−^ > S_2_O_3_^2−^ > H_2_PO_4_^−^ > F^−^ > Cl^−^ > Br^−^ ~ NO_3_^−^ > I^−^ > ClO_4_^−^ > SCN^−^. The left species of Cl^−^ are called kosmotropes, while the ones on the right are referred to as chaotropes. In fact, changes in protein hydration induced by salts are one of the driving forces of liquid–liquid phase separation [20], which plays a critical role in the health and disease states of living organisms. Later, the Hofmeister series were used to explain the thermal-responsive behaviors of polymers in aqueous solutions based on the direct interactions of anions with macromolecules [21,22,23,24,25]. For example, Cremer and coworkers systematically studied specific ion effects on the basis of the Hofmeister Series on the thermal-responsive behavior of PNIPAM [26]. This has inspired tremendous efforts for the fabrication of smart polymeric materials with modulated thermal-responsive properties [27,28,29,30,31,32,33,34].

Herein, a new type of thermal-responsive polyacrylamide-based block-copolymer (Px) bearing pyridine groups was designed and synthesized through reversible addition-fragment chain transfer (RAFT) polymerization, showing unique salt-induced LCST behavior in aqueous solution. This thermo-responsive polymer bearing pyridine-containing acrylamide groups totally differs from traditional poly(acrylamide) analogs such as PNIPAM in terms of its chemical structure and responsive mechanisms. The significant influence on the thermal behaviors of the block copolymers exerted by molecular weight, salt concentration and species and pH values was systematically demonstrated. In addition, the obtained block copolymers were able to self-assemble into polymeric nano-objects in an aqueous solution—driven by the complementary hydrogen bonds in the *N*-(2-methylpyridine)-acrylamide (MPA)—that possess pH-responsive properties due to the protonation and deprotonation of their pyridine group. It can be anticipated that there will be a new class of thermo-responsive block polymers with pH responsive properties that can self-assemble into polymeric nanostructures driven by hydrogen bonding interactions.

## 2. Results and Discussion

### 2.1. Synthesis and Characterization of Diblock Copolymers (Px) via RAFT Polymerization

Px was synthesized via heat-initiated RAFT polymerization by employing macro-CTA (PEG-CTA) in the presence of MPA in ethanol at 70 °C for 24 h (Figure 1a). By controlling the molar ratio of the MPA and PEG-CTA, a series of block copolymers with different levels of PMPA were obtained. The GPC analysis showed that after polymerization, the molecular weight of the Px with a narrow index of polymer distribution (PDI < 1.3) increased with increases in the molar ratio of the MPA and PEG-CTA (Figure 1b). This suggests that the polymerization was controllable. Subsequently, the molecular structure of the Px was determined by proton nuclear magnetic resonance (^1^H NMR) and Fourier transform infrared spectrometer (FT-IR) after being purified by dialyzing in water, using a molecular cut-off 3.5 kDa. The ^1^H NMR spectrum exhibited typical pyridine peaks for PMPA at 8.4–7.1 ppm, and for PEG located at ca. 3.5 ppm (Figure 1c and Appendix A). The FT-IR spectrum of the PEG-PMPA displayed representative signals at the 1661 and 1566 cm^−1^ wavelengths (Figure 1d), which illustrated the stretching vibration absorbance of the amido bonds and pyridine rings, respectively, and at 1103 cm^−1^—demonstrating the ether bond stretching vibration absorbance of the PEG backbone segments. These results confirmed the obtaining of PEG-PMPA via controllable RAFT polymerization.

### 2.2. Salt-Induced Thermal Response of Px

After obtaining the polymers, we first studied the different effects of salts, chemical additives and their concentrations on the aqueous thermal behaviors of the Px block copolymers. It has been shown that ionic strength is capable of influencing the solubility of macromolecules with amide groups by influencing hydrogen bonding between amide groups and water [26]. Thus, salt’s effects on the solubility of the obtained Px block copolymers were investigated. As homopolymers of PMPA are hardly homogeneously dispersed in water—even with a low DP (degree of polymerization) of only 20 (Appendix A)—purified block copolymers of Px with different DPs (17: P1, 21: P2, 30: P3, 38: P4) that are able to disperse in water after ultrasound for ca. 10 min at room temperature were selected. When continuously adding NaCl from 100 mM to 1.5 M into the solution, the ^1^H NMR spectra illustrated that the signals of the pyridine peaks increasingly weakened and became broad (Appendix A), which suggested that a high ionic strength could reduce the solubility of P2.

Next, temperature’s effect on the solubility of Px was systematically studied. As shown in Figure 2a (black line), turbidity experiments revealed that little transmittance (at 500 nm) change was observed with temperature increases from 20–75 °C in deionized water, which illustrated that no obvious aggregation occurred during heating in deionized water. Furthermore, the temperature-variable ^1^H NMR (TV-NMR) spectra displayed that, when enhancing the temperature from 25 to 60 °C, the proton signal peaks of the pyridine ring gradually became slightly sharpened and narrowed, following a gradual increase in the relative integrated value of the protons in the pyridine (Figure 2d, purple line). This can be ascribed to the fact that high temperatures, to a certain degree, weaken the inter-molecular hydrogen bonds between pyridine and amide bonds—increasing the water solubility of the polymer. It is well-known that the thermal properties of acetamide-based polymers such as PNIPAM can be influenced by anions based on the Hofmeister series in aqueous solutions [35]. Thus, we studied salt’s effects on the possible thermal behaviors of Px in water. Intriguingly, when we added even a small amount of NaCl (30 mM) into the P2 system (1 mg/mL), a cloud-point temperature (CPT) of 66 °C appeared, and when the concentration of NaCl was increased from 30 mM to 1 M, the CPT decreased from 66 °C to 47 °C (Figure 2a). Meanwhile, the TV-NMR analysis revealed that when increasing the temperature from 25 to 60 °C, with a concentration of NaCl of 700 mM in D_2_O, the proton peaks of the pyridine rings gradually weakened and became broad (Figure 2c), following an evident decrease in the relative integrated value of the protons in the pyridine (Figure 2d, black line)—which agrees with the turbidity results. Besides this, the effects of other anions including kosmotropes and chaotropes on the thermal behaviors of Px were also investigated. The results in Figure 2b and Appendix A exhibit that all the selected anions, regardless of the presence of kosmotropes (Br^−^ and I^−^) or chaotropes (SO_4_^2−^), were able to induce the appearance of CPT. Furthermore, all the cloud-point temperatures of Px decreased with increases in the concentrations of NaCl, NaBr, NaI and Na_2_SO_4_. Additionally, the order-of-change range induced by the addition of salts was Na_2_SO_4_ >> NaI > NaBr ~ NaCl—indicating that the thermal response of Px influenced by anions did not follow the classical Hofmeister serial order, which differs from that of traditional synthesized temperature-responsive polymers such as PNIPAM or the family of poly (oligo(ethylene glycol)). This unique phenomenon may be mainly attributed to the electron-rich nitrogen present in pyridine structures that forms complementary hydrogen bonds with amide groups. The control experiment with the PEG-CTA and MPA monomer showed no turbidity variation with temperature increases up to 1 M NaCl concentrations (Appendix A), excluding the PEG block and a possible monomer residue effect on the salt-induced LCST. From all these results, it may be deduced that salts can significantly influence the solubility and unique thermal behaviors of Px block polymers—possibly via the polarization of an adjacent water molecule, weakening the hydrogen bonding interactions between water molecules and amides and, in contrast, enhancing the complementary hydrogen bonding between pyridines and amides.

Subsequently, aggregation behavior was observed upon heating. As shown in Figure 3a, it was found that the P2 formed irregular, loose aggregates at 25 °C with NaCl concentrations of 500 mM, driven by complementary hydrogen bonding between the pyridine and amides—subsequently assembled into large micelles with a size of several micrometers (Figure 3b). The temperature-variable DLS (TV-DLS) results showed a significant size increase with temperature enhancement at different NaCl and Na_2_SO_4_ concentrations (Figure 3c,d). Intriguingly, this thermal response was totally responsive (Appendix A).

In order to demonstrate the key role of hydrogen-bonding interactions between the pyridine and amides in the temperature response of the Px, varying amounts of urea—which is commonly utilized to disturb hydrogen bonds—were added into the solution [36]. It was observed that with increases in urea concentrations from 0 to 7 M, the CPT of the P2 (1 mg/mL, 500 mM NaCl) significantly increased from ca. 45 to 70 °C, and when adding urea at over 8 M, no obvious temperature response was observed (Figure 4)—suggesting that urea could dramatically enhance the aqueous solubility of Px.

After verifying the salt effect on the temperature properties of the Px, we next studied the effect of DP and its concentration on the thermal response of Px. As illustrated in Appendix A, four kinds of Px with a PMPA DP of 17, 21, 30 and 38, respectively, were evaluated at various concentrations (0.5 mg/mL, 1 mg/mL and 2 mg/mL) and 500 mM NaCl. Generally, with increases in both the DP and its concentration, the cloud-point temperature shifted to a lower temperature. Temperature-dependent turbidity curves exhibited that the cloud-point temperature shifted from 61 to 41 °C (0.5 mg/mL), 59 to 37 °C (1 mg/mL) and 57 to 31 °C (2 mg/mL) with increases in the DP of the PMPA (Appendix A)—possibly due to the increased efficiency of intermolecular aggregation with molecular weight increases. Notably, when the DP of the PMPA part was 38, it was found that the transmittance was lowered to 27 % (2 mg/mL) at 20 °C, which implies the formation of large aggregates from P4 at a DP of 38. Subsequently, the temperature-responsive behavior of Px was investigated at different concentrations. It was observed that when increasing the concentration from 0.5 to 2 mg/mL at NaCl 500 mM, the cloud-point temperature decreased from 61 to 57 for DP 17, 54 to 50 for DP 21, 44 to 39 for DP 30 and 41 to 31 for DP 38 (Appendix A), which could be attributed to the increased efficiency of the aggregation of intermolecular chains with increasing concentrations.

### 2.3. pH-Responsive Properties of Px

Considering that pyridine is structurally sensitive to acid, the pH effect on the obtained Px polymers was investigated. It can be anticipated that the pH value can be used to not only tune the solubility of Px, but also to produce an effect on their thermal behaviors in aqueous solution. The ^1^H NMR spectra (Figure 5a) showed that after adding 0.2 M DCl (deuterium chloride) into the purified P3 (10 mg/mL) D_2_O solution, all the signals of the PMPA block became dramatically sharp and strong, manifesting that an acid can significantly enhance the solubility of the block copolymers (Px) due to the protonation of their pyridine groups. Furthermore, we systematically investigated pH effects on the temperature-responsive behaviors of P3 in PBS buffer solution containing a certain amount of salt (ionic strength ≈ 0.5 mol/L). It was found that when varying the pH from 9.0 to 3.0, the CPT of the P3 shifted from 33 to 57 °C (Figure 5b,c), which was ascribed to the constant protonation that occurred with pH decreases. However, when lowering the pH to 2, the CPT disappeared (Figure 5b)—possibly due to the fact that the dense protonation of pyridine moieties completely prevents the aggregation of PMPA blocks. In addition, their CPT values can be reversibly regulated by alternatively changing the pH values (Figure 5d). All these results indicate that the resultant Px block copolymers possessed a remarkable pH-responsive feature due to the protonation and deprotonation of their pyridine groups. Furthermore, at the same time, the CPT of the polymer could be precisely tuned within a large range by altering the pH in buffer solution.

## 3. Materials and Methods

### 3.1. Materials

Acryloyl chloride (Aladdin, 98%), 4-Dimethylaminopyridine (Aladdin, 98%), Methoxypolyethylene glycols (2000; Aladdin, 98%), 2-Picolylamine (Aladdin, 99%), 2-Butylsulfanyl-thiocarbonylsulfanyl-propionic acid (Aladdin, 98%), Triethylamine (Aladdin, 98%), Azobisisobutyronitrile (Aladdin, 98%) and 1-(3-Dimethylaminopropyl)-3-ethylcarbodiimide hydrochloride (Aladdin, 98%) were purchased from Aladdin Reagents Co. Ltd. (Shanghai, China).

### 3.2. Characterizations

NMR spectra were obtained from Agilent DD2 600 MHz of Bruker BioSpin International (Billerica, MA, USA). CDCl_3_ and D_2_O were used as solvents. The structures of the nanomaterials were observed by transmission electron microscopy (TEM, HT-7700 microscope, Hitachi, Japan). Dynamic light scattering (DLS) was performed using a Delsa Max Pro from Beckman Coulter (Brea, CA, USA). UV-Vis Absorption Spectra: UV-vis spectra were recorded on an Analytikjena Specord S 600 (Jena, Germany) via using a cuvette to scan at 200–800 nm. FT-IR spectra were collected using a Nicolei 6670 FT-IR spectrometer (Madison, WI, USA).

### 3.3. Synthesis of Px

MPA, PEG-CTA and AIBN with specific molar ratios were added into 1 mL ethanol. After 3 cycles of freezing and thawing, the reaction was kept at 70 °C for 24 h to ensure that conversion approached 100%. MPA and PEG-CTA were synthesized according to our previous work [36].

## 4. Conclusions

In summary, we synthesized a new type of polyacrylamide-based block copolymer bearing pyridine groups that showed salt-induced LCST behavior. The LCST can be tuned by several external factors such as salt species and concentration, pH value, urea concentration, molecular concentration and the DP of the PMPA block. It is notable that the mechanisms of the thermo-responsive behaviors of the PMPA block did not strictly follow the order of the Hofmeister series, showing different mechanisms to those of traditional LCST-possessed polymers. We believe that this novel kind of thermal-responsive block-copolymer and its assemblies possess great potential in a wide range of applications.

## Figures and Tables

**Figure 1 molecules-28-02921-f001:**
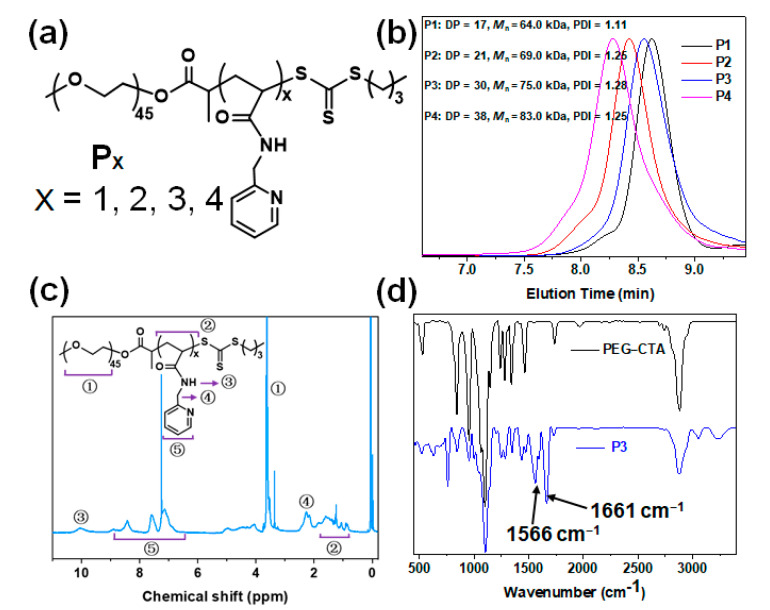
(**a**) Chemical structures of the block-copolymers P1–P4. (**b**) GPC curves of P1–P4 after the RAFT polymerization of MPA in ethanol. (**c**) Typical ^1^H NMR spectrum of P3 in CDCl_3_ at 25 °C. (**d**) Typical FT–IR spectra of P3 (blue line) and PEG–CTA (black line).

**Figure 2 molecules-28-02921-f002:**
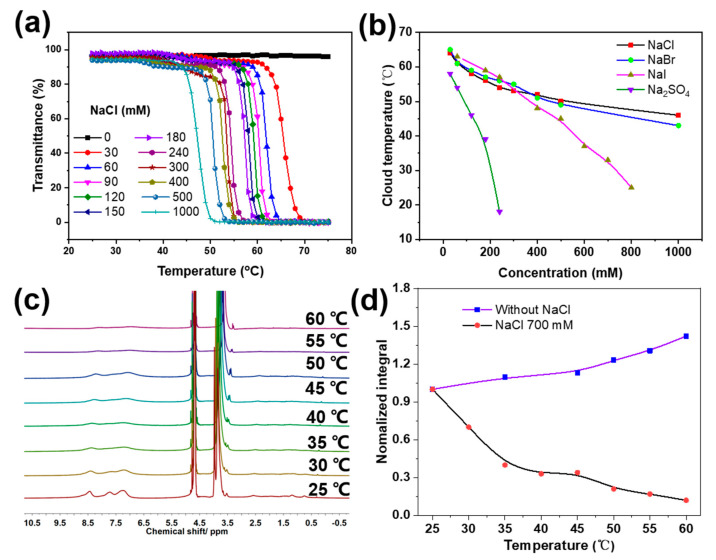
(**a**) Effect of NaCl concentration on the thermal response of P2. (**b**) Effect of various salts on the temperature–responsive behaviors of P2. (**c**) ^1^H NMR spectra of P2 at various temperatures with NaCl concentrations at 700 mM. (**d**) Plots of the integration area of the protons on the pyridine ring versus temperature with (black line) and without (purple line) NaCl.

**Figure 3 molecules-28-02921-f003:**
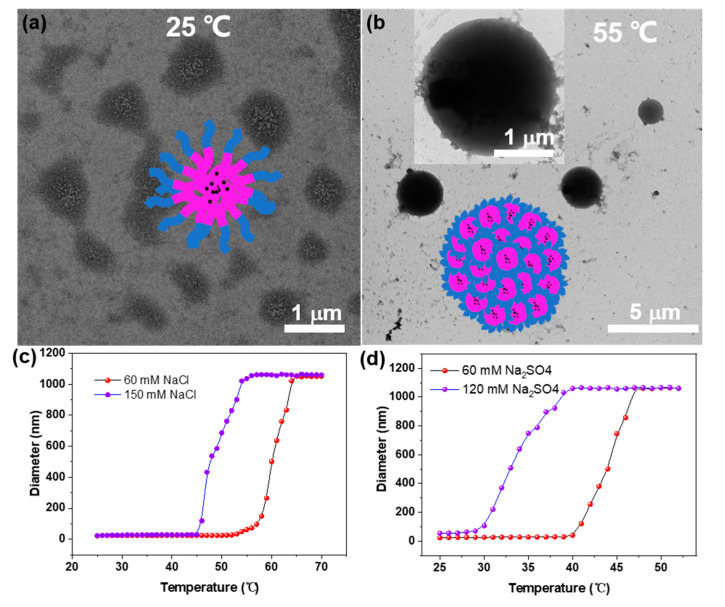
TEM images of the P2 at (**a**) 25 °C and (**b**) 55 °C with 500 mM NaCl. The diameter variation of P2 upon the temperature increases with different concentrations of (**c**) NaCl and (**d**) Na_2_SO_4_.

**Figure 4 molecules-28-02921-f004:**
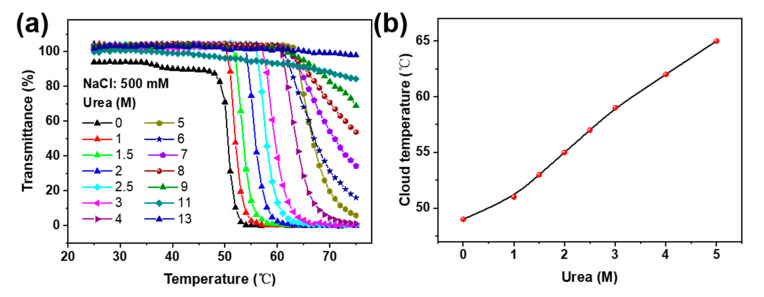
(**a**) Effect of urea concentration on the thermal response of P2 in NaCl aqueous solution. (**b**) Plots of the cloud-point temperature versus urea concentration in NaCl solution.

**Figure 5 molecules-28-02921-f005:**
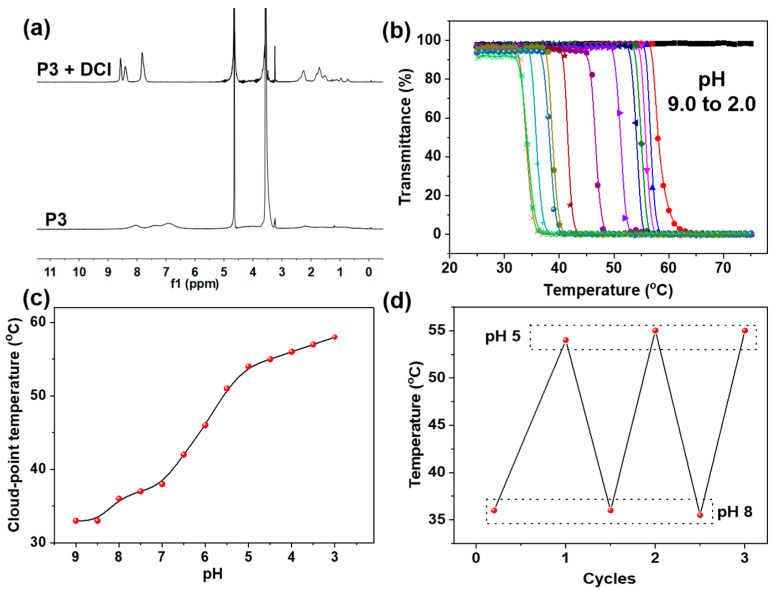
(**a**) ^1^H NMR spectra of P3 before and after the addition of DCl in D_2_O. (**b**) Effect of pH values on the thermal-responsive behaviors of P3 in PBS buffer. (**c**) Plots of the cloud-point temperature depending on the pH values in PBS buffer. (**d**) Reversible cycles of cloud-point temperature by alternatively changing the pH between 8 and 5.

## Data Availability

The data presented in this work are available in the article and Appendix A.

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
