# Peer review of "Polyacrylamide-Based Block Copolymer Bearing Pyridine Groups Shows Unexpected Salt-Induced LCST Behavior"

_molecules, 2023, doi:10.3390/molecules28072921_

Round 1

Reviewer 1 Report

In this report, Yang et al. synthesized a new blocky copolymer featuring both PEG and polyacrylamide blocks, that displayed interesting LCST performances, as revealed from salt- and PH- responsive behaviors. The studies is routine, but the characterizations is sound, which convinces me to accept this paper, but a major revision is required. The authors should pay attention to the following issues, and correct them accordingly.

1.    Instead of FTIR, 1H and 13C NMR spectra should also provided for all the obtained polymers, because only NMR can give a quantitative calculations.

2.    The order of a-d in Figure 1 is wrong, please correct.

3.    Why the trend of NaI, NaBr, NaCl differed from previous reports? Please give reasonable reasons.

4.    Grammar issues:

(1)   page 3, “the 1H NMR spectra illustrated that…………and become broad, which suggests that ………” should be “became broad, which suggested that……”

(2)   page 3, “no obvious aggregation occurs during………” should be “occurred”.

(3)   Page 4, “concentration increasing of NaCl” should be” concentration of NaCl increasing”

Reviewer 2 Report

In this submission, the author reported a new type of poly-acrylamide-based block copolymer bearing pyridine groups of polyethylene glycol-block-poly(N-(2-methylpyridine)-acrylamide), displayed a distinct salt-induced LCST. The author reported that these block-copolymers and their assemblies have a bright future in various biomedical applications.

The manuscript has been well written, except for a few errors, and could be published after revision.

 Here are the comments which need to be addressed.

1.     Figure 1a: The letter 'n' has been used to designate the sample name. However, 'n' has also been used to mention the degree of polymerization. It would be better to change the sample designation to increase the reader’s understanding.

2.     In the Figure 1C: please correct the molecular weight presentation as 'kDa', not the 'kD'

3.     "The GPC analysis shows that after polymerization, the molecular weight of Pn with a narrow index of polymer distribution (PDI<1.3) increases via increasing the molar ratio of MPA and PEG-CTA (Figure 1b). "

The mentioned figure 1b should be figure 1c. Please correct it throughout the manuscript. Figures 1b and 1c might be switched between. Please correct it accordingly.

 4.     The author needs to define 'DP' when it appears first in the manuscript.

How has the author calculated the DP for all the mentioned copolymers? For example, P1, 21: P2, 30: P3, 38: P4”

Also, it would be good to define the 'DCl'.

5.     The author could find “Tailor-Made Temperature-Sensitive Micelle for Targeted and On-Demand Release of Anticancer Drugs” a relevant article on LCST-based copolymers to include in the introduction section.

Round 2

Reviewer 1 Report

it can be published in the present form.